# Paliperidone–Cation Exchange Resin Complexes of Different Particle Sizes for Controlled Release

**DOI:** 10.3390/pharmaceutics15030932

**Published:** 2023-03-13

**Authors:** Jun-Pil Jee, Young Hoon Kim, Jun Hak Lee, Kyoung Ah Min, Dong-Jin Jang, Sung Giu Jin, Kwan Hyung Cho

**Affiliations:** 1College of Pharmacy, Chosun University, Gwangju 61452, Republic of Korea; 2College of Pharmacy and Inje Institute of Pharmaceutical Sciences and Research, Inje University, Gimhae 50834, Republic of Korea; 3Department of Bio-Health Technology, College of Biomedical Science, Kangwon National University, Chuncheon 24341, Republic of Korea; 4Department of Pharmaceutical Engineering, Dankook University, Cheonan 31116, Republic of Korea

**Keywords:** cation exchange resin, paliperidone, resin particle size, controlled release

## Abstract

This study aimed to develop electrolyte complexes of paliperidone (PPD) with various particle sizes using cation-exchange resins (CERs) to enable controlled release (both immediate and sustained release). CERs of specific particle size ranges were obtained by sieving commercial products. PPD–CER complexes (PCCs) were prepared in an acidic solution of pH 1.2 and demonstrated a high binding efficiency (>99.0%). PCCs were prepared with CERs of various particle sizes (on average, 100, 150, and 400 μm) at the weight ratio of PPD to CER (1:2 and 1:4). Physicochemical characterization studies such as Fourier-transform infrared spectroscopy, differential scanning calorimetry, powder X-ray diffraction, and scanning electron microscopy between PCCs (1:4) and physical mixtures confirmed PCC formation. In the drug release test, PPD alone experienced a complete drug release from PCC of >85% within 60 min and 120 min in pH 1.2 and pH 6.8 buffer solutions, respectively. Alternatively, PCC (1:4) prepared with CER (150 μm) formed spherical particles and showed an almost negligible release of PPD in pH 1.2 buffer (<10%, 2 h) while controlling the release in pH 6.8 buffer (>75%, 24 h). The release rate of PPD from PCCs was reduced with the increase in CER particle size and CER ratio. The PCCs explored in this study could be a promising technology for controlling the release of PPD in a variety of methods.

## 1. Introduction

Schizophrenia is a neuropsychiatric disorder that affects 1% of the world’s population [1]. Its treatment mainly employs both typical and atypical antipsychotics due to their relatively tolerable profiles and broad clinical activity. Paliperidone (PPD, Figure 1A), or 9-hydroxyrisperidone, is the major active metabolite of the atypical antipsychotic risperidone and is one of the most recent antipsychotic medications on the market and is widely prescribed [2]. PPD possesses pharmaceutical differences compared to risperidone in terms of its predominant renal metabolism, lower protein binding, and decreased inhibition of P-glycoprotein, leading to a decreased potential for drug–drug interactions [3]. As with other drugs having efficacy in schizophrenia, the mechanism of action of this second-generation antipsychotic, PPD, is unknown. However, it has been proposed that the drug’s therapeutic activity in schizophrenia is mediated through a combination of central dopamine D2 receptors in the mesolimbic pathway and 5-HT2A receptors in the prefrontal cortex [4].

Schizophrenia patients are often required to adhere to complex medication schedules involving multiple daily doses to maintain optimal plasma drug levels. This may be difficult, particularly for disorganized patients [5]. Thus, PPD was approved for an extended-release (ER) tablet (Invega^®^, Janssen, Antwerp, Belgium) for a once-daily administration based on an osmotic-controlled release oral Push–Pull™ delivery system (Oral Osmotic System, OROS^®^, Alza Corporation, Mountain View, CA, USA) [6]. In addition, studies on liquid suspensions using ion exchange resin, oral formulation using lipid nanostructures, and injections using liposome systems have been conducted for the controlled release of PPD [7,8,9].

A controlled release system is used to release a drug at an intended rate over an intended time and has enhanced therapeutic effects via the controlled plasma concentration of the drug [10]. Controlled-release technologies can be broadly classified into pump systems and polymer systems. Controlling drug release using a pumping device is the most direct approach. However, a pump system is much more expensive and can cause the unwanted problem of dose dumping if the pump breaks [11]. Polymer controlled-release systems that use biodegradable and non-biodegradable, soluble and insoluble polymers can be administered as drug carriers via various administration routes [12,13,14]. Among the controlled-release technologies, drug–resin complexes that use ion-exchange resins are one of the leading applications in the pharmaceutical industry because they are simple, cost-effective, and do not require more components or organic solvents [15]. In this study, controlled release aims to implement a system that can control the release of drugs quickly or slowly without applying special techniques using ion exchange resins.

Ion-exchange resins are water-insoluble synthetic polymers containing acidic or basic functional groups in a repeating pattern. They are widely used to release free drugs by forming reversible weak ionic bonds with drugs based on charge interactions [16]. When charged ions (sodium, potassium, or chlorine ions) replace the drug on the ion-exchange resin in the digestive tract, the drug is released [17]. In the past, ion-exchange resins were mainly used in agriculture and water purification. In the field of medicine, ion-exchange resins have been extensively studied for their potential use in drug delivery systems and are key materials in pharmaceutical development due to their unique properties, such as stabilizing sensitive drugs, improving unpleasant taste, and controlling drug release [18,19]. Ion-exchange resins are promising carriers of many ionized drugs because of their good physicochemical stability and lack of local or systemic side effects. This could help solve problems with drug delivery. Furthermore, drug complexes with ion-exchange resins are stable, easily manufactured, and cost-effective [20]. In addition, since the release pattern of the complex depends on the chemical process of ion exchange, these drugs are suitable for controlling drug release [21]. However, few studies have been conducted on the use of controlled release according to various physicochemical properties, such as particle size. In the development of PPD and ion-exchange resin complexes, the particle size of ion-exchange resins is very important in drug binding and release.

Ion-exchange resins can be divided into two classes: cation- and anion-exchange resins. Cation-exchange resins (CERs; Figure 1B) are suitable as carriers for cationic drugs, and the functional group of CERs used in pharmaceuticals is mainly carboxylic acid (-COOH) or sulfonic acid (-SO_3_H) [22]. The degree of their dissociation is strongly influenced by the solution pH. In particular, as a representative CER, Amberlite^TM^ IRP 64 has a cation exchange capacity of about 10 meq/g (milliequivalents per gram of dry resin, amount of exchangeable ions) and is derived from a porous copolymer of methacrylic acid and divinylbenzene [23].

PPD has a piperadine moiety in its structure (Figure 1A) that acts as a basicity and is protonated in an acidic solution. PPD was reported to form an electrolyte complex with dextran sulfate polymer, which has a structure containing sulfonate with a strong negative charge. In this study, PPD was focused on the possible ormation of a complex with CER having a sulfonate group, and the CER properties such as particle size were considered to be a critical variable for controlling drug release [24]. In addition, the development of a controlled-release oral system via a simple manufacturing method for PCC was of great significance to schizophrenia patients, providing a new alternative pharmaceutical treatment modality.

This study aimed to establish conditions that control the formation ability and release characteristics of a PPD–CER complex (PPC) according to the difference in the particle size of CER and to confirm the potential of this complex for using a carrier for controlled release. PCC was obtained in a batch method in two weight ratios using CERs (AmberLite^TM^ IRP69, AmberLite^TM^ IR69-F, DIAION^TM^ UBK530) with different particle size distributions, classified by standard sieves. After preparation, the physicochemical properties of PCC were studied through binding efficiency, Fourier-transform infrared spectroscopy (FT-IR), differential scanning calorimetry (DSC), powder X-ray diffraction (PXRD), and scanning electron microscopy (SEM) evaluation. A drug release test for the prepared PCCs was performed in pH 1.2 and pH 6.8 buffers to investigate the properties of controlled release.

## 2. Materials and Methods

### 2.1. Materials

PPD was purchased from Hangzhou Hyper Chemicals Limited (Hangzhou, China). AmberLite^TM^ IRP69 (Sodium Polystyrene Sulfonate) and AmberLite^TM^ IR69F (Sodium Sulfonated Poly(Styrene-Co-Divinylbenzene)) were provided by Colorcon Asia Pacific Pte. Ltd., Korea Branch (Suwon, Republic of Korea). DIAION^TM^ UBK530 (Sodium sulfonated poly(styrene-co-divinylbenzene)) was provided by Samyang Co. (Seoul, Republic of Korea). All other chemicals were of reagent grade and were used without further purification.

### 2.2. HPLC Conditions

The PPD amount in samples was analyzed using a Waters 2695 HPLC system (Waters, Milford, MA, USA) equipped with a UV–Vis detector (Waters 2487; Waters, Milford, MA, USA). PPD was separated on a reverse-phase C18 column (250 mm × 4.6 mm, 5 µm, Shiseido, Tokyo, Japan) at a flow rate of 1.0 mL/min. The mobile phase was a mixture of ammonium acetate buffer (pH 4.0) and acetonitrile (50:50, *v*/*v*), filtered through a 0.45-micron nylon filter. The injected volume of the sample was 10 μL, and UV detection was performed at 235 nm. Data acquisition and processing were performed using the Waters LC Solution software (Empower 2.0 version).

### 2.3. Preparation of PCC

PCCs were prepared with PPD and CER via a batch method [25]. Before use, each 100 g CER was purified by stirring for 3 h at 500 rpm with 200 mL of 0.1 N HCL, followed by filtering using a vacuum filtration device and drying in an oven at 50 °C for 24 h. Then, each CER fraction of C1 (AmberLite^TM^ IRP69), C2 (DIAION^TM^ UBK530), and C3 (AmberLite^TM^ IR69F) was prepared by the following method using standard sieves: All the sieving below was conducted with 30 g of CER and 1 h of sieve shaking time. Standard sieves of #100, #200, and #325 mesh were used for C1, and only the portion remaining on the #200 mesh was obtained. For C2, only the portion remaining on the #50 mesh was obtained after sieving with #40, #50, and #60 mesh. Finally, for C3, #30, #40, and #50 mesh were used, and only the remaining portion of #40 mesh was obtained. C1, C2, and C3 were dried in a silica desiccator for 24 h. PCCs were prepared using two weight ratios (1:2 and 1:4, *w*/*w*) of PPD to C1, C2, and C3, respectively. After completely dissolving 600 mg PPD in 24 mL of 0.1 N HCl, 1.2 or 2.4 g of C1, C2, and C3 were added and stirred at 300 rpm for 24 h. The mixture was filtered with a 0.45 μm membrane filter and rinsed with 6 mL of water, and the remaining material was dried in an oven at 50 °C for 24 h, and PCCs were obtained. The physical mixtures were prepared by geometric mixing in the same proportions as each PCC composition.

The PPD binding efficiency (%) in PCC was calculated by analyzing the concentration of unbound drug in the filtrate (total of 30 mL). The binding efficiency (%) of PPD–CER in PCC was calculated via the following equations [26].
$$\mathrm{Binding}\;\mathrm{efficiency}\;\left(\%\right)=\left(\frac{600\;\mathrm{mg}-\left(\mathrm{Concentration}\;\mathrm{of}\;\mathrm{PPD}\;\mathrm{in}\;\mathrm{filtrate}\;\left(\mathrm{mg}/\mathrm{mL}\right)\times 30\;\mathrm{mL}\right)}{600\;\mathrm{mg}}\right)\times 100\;\left(\%\right)$$

### 2.4. Fourier Transform Infrared Spectroscopy (FT-IR) Characterization

FT-IR measurements were performed to characterize the interaction details between PPD and CER. The transmission infrared spectra of samples were recorded using an FT-IR spectrophotometer (Varian 640-IR; Palo Alto, CA, USA). The spectral width was 4000 cm^−1^~500 cm^−1^, samples were scanned 16 times, and all samples were measured with the attenuated total reflectance method [27].

### 2.5. Differential Scanning Calorimetry (DSC) Characterization

DSC characterization was performed to characterize the physical state of PPD in samples. DSC thermograms were recorded with a DSC instrument (TA Instruments; New Castle, DE, USA). A sample of approximately 10 mg was weighed into an aluminium pan with a sealed lid. Measurements were performed under a nitrogen purge over 10~250 °C with a heating rate of 10 °C/min [28].

### 2.6. Powder X-ray Diffraction (PXRD) Characterization

A PXRD characterization of the sample was recorded using an X-ray diffractometer (Rigaku IV Ultima; Tokyo, Japan), which was equipped with a Linxeye 1-D detector. Each sample was added to the grid, and the diffraction pattern of each sample was measured using a Cu Kα radiation source (40 kV and 40 mA) with an acquisition time of 0.2 s per step. The scanning range was 5°~40° in the 2θ range [29]. 

### 2.7. Scanning Electron Microscope (SEM) Analysis

The morphologies of samples were investigated using SEM. The dried samples were affixed to the specimen stubs using double-sided copper tape sputter-coated with palladium in the presence of argon gas using an ion sputter coater (Hitachi E-1030; Tokyo, Japan). Samples were imaged on a scanning electron microscope (Hitachi S-4300; Tokyo, Japan) using a 15 kV accelerating voltage [30].

### 2.8. Particle Size Measurement

The particle size of C1~C3 was measured by the Mastersizer 3000 (Malvern, Worcestershire, UK) in dry powder form. The instrument was optimized under the following conditions: air pressure, 1 bar; feed rate, 50%; gab, 1 [31].

### 2.9. In Vitro Release Test of PCCs

Each sample was filled into a gelatin capsule (#0), equivalent to 9 mg PPD. In the case of the PCC mixture, PCC1, PCC2, and PCC6 were mixed in the ratio of 10:60:30 (*w*/*w*) to be 9 mg of PPD in a capsule. Drug release tests were conducted according to USP Apparatus 2 guidelines (paddle method) (Varian VK 7000: Cary, NC, USA) with 900 mL of dissolution medium maintained at 37 ± 0.5 °C and mixed at 50 rpm. The dissolution media used in this study were pH 1.2 and pH 6.8 buffers. Aeach time interval of 5, 10, 15, 30, 45, 60, and 120 min at pH 1.2; 5, 10, 15, 30, 45, 60, 120, 240, 360, 480, 720, 1080, and 1440 min at pH 6.8; 3 mL of sample was withdrawn and immediately replenished with an equal volume of fresh dissolution medium. A 1 mL sample was taken, centrifuged, and the amount of dissolved drug was analyzed by HPLC as mentioned above [32].

### 2.10. Statistical Analysis 

A statistically significant difference was confirmed by the Student’s *t*-test (for a pair of groups) and one-way ANOVA followed by Tukey’s post hoc test (for more than two groups). For the statistical calculation, SPSS Version 26 (IBM, Armonk, NY, USA) was utilized. A *p*-value of <0.05 was considered significant. 

## 3. Results and Discussion

### 3.1. Physicochemical Characterization of the CER and PCC

PCC manufacturing methods can be largely classified into batch or column methods [15]. A common batch method is to mix a drug solution with an ion-exchange resin, and a column method is prepared by eluting a concentrated drug solution through a column filled with an ion-exchange resin. For both methods, the ion-exchange resin complex that has reached a state of equilibrium is washed and dried to get rid of any unbound drugs. In this study, PCC was prepared using the batch method. Moreover, the CERs (AmberLite^TM^ IRP69, DIAION^TM^ UBK530, and AmberLite^TM^ IRP69F) in this study possess the structural sulfonate functional group, which is almost completely ionized regardless of solution pH. The pKa of piperadine in PPD was 8.3 and it was ionizable in 0.1 N HCl solution (pH 1.2) used for PCC formation [26,33,34].

PPD, a weakly basic drug with a piperadine moiety, had poor solubility in water and a neutral pH, was sparingly soluble in 0.1N HCl, and had pH-dependent solubility. The solubility in water was 30 μg/mL and increased in an acidic solution [35]. Thus, the actual solubility in 0.1N HCl was evaluated to select the concentration of PPD needed for the preparation of PCC. PPD showed solubility of >30 mg/mL in this study (specific data not shown). Therefore, when preparing PCC, the concentration of the PPD solution was set to be a lower 25 mg/mL for easy dissolution. In the particle size measurement of CERs, the averages for C1, C2, and C3 were 105.5 ± 7.6, 153.7 ± 17.2, and 407.6 ± 20.1 μm, respectively (Table 1). The particle size distributions were uniform, with a low relative standard deviation of <20% due to the sieving treatment classification. In addition, each CER exhibited the shape of a powder flake (C1), a spherical bead (C2), and a spherical bead (C3), which originated from commercial production.

To determine the effect of the ratio of PPD over CER on the binding efficiency (%) of PPD, PCC was prepared in two ratios (PPD:CER = 1:2 and 1:4), respectively. The binding efficiency of PPD in PCC was summarized in Table 2. The PCC1~PCC4 using C1 and C2 had a significantly higher binding efficiency of >98% than PCC5. On the other hand, the PCC5 and PCC6 using C3 had average binding efficiencies of 62.10% and 98.31%, respectively. In the case of C3, the binding efficiency (%) increased as the ratio of CER increased. In the cases of C1 and C2, a sufficient surface area of CER for PPD during the preparation of the PCC secured a very high binding efficiency (%) at both ratios of 1:2 and 1:4. However, in C3, a slightly lower binding efficiency was due to its relatively large particle size and small specific surface area [36]. Furthermore, when PPD molecules react with larger CER particles, they are more likely to diffuse into the interior of the particles. This reduces mass transfer and makes most of the inner resin bead less accessible for binding. This results in a lower binding efficiency compared to smaller CER particles with larger specific surface areas [26]. Among the CERs used, C1 and C2 had the most desirable properties for the binding efficiency of PPD. When using a limited amount of CER relative to the drug, the particle size of the CER affected the binding ability of PPD and CER.

The particles were analyzed using SEM to observe the surfaces of PPD, CER, PCC2, PCC4, and PCC6 (Figure 2). PPD appears as a flaky structure about 30 μm in size, suggesting the crystalline nature of the PPD [35]. In the case of the CERs (C1, C2, and C3), they were about 100–130 μm, 140–170 μm, and 400–430 μm, respectively (Figure 2B,E,H). This was found to be consistent with the particle size range and particle size analysis results, as shown in Table 1. In the physical mixture (Figure 2C,F,I), PPD particles were attached to larger CER particles due to electrostatic forces. The surface of PCC2 prepared with C1 had a lubricity (similar to a film coating) on the surface compared to CER alone (Figure 2B,D). The particle size of the PCC4 and PCC6 with a bead-like appearance (Figure 2G,J) increased by about 30~50 μm on average compared to each single CER, C2, and C3. This significant increase in PCC particle size means that the PPD molecules should diffuse into the inner CER to form electrostatic bonding rather than simple surface bonding. As a result, the crystal structure of PPD was not shown in the prepared PCCs, and the particle size and surface characteristics in PCCs were changed from CERs [17,26].

The formation of PCC was evaluated using an FT-IR analysis (Figure 3). In the comparison of raw PPD, PCC, and CER (C1, C2, and C3), and each physical mixture, the specific peak band derived from the PPD or CER disappeared or changed in the PCC. In the raw PPD and the physical mixture of PPD and CER, characteristic peak bands of PPD were observed at 2800~3000 cm^−1^ and 3200~3400 cm^−1^. However, in PCC, these two peaks were not present, as indicated by the red-dotted square in Figure 3 [37]. Similar to previous studies, the peak band at 2200~2400 cm^−1^ was weakened in PCC compared to the higher intensity peak band in the physical mixture [22,38]. This result means the existence of structural interaction between PPD and CER, which is expected due to the electrostatic coupling of piperadine from PPD and sulfonate from CER [39]. However, for exploring more evidence to confirm the complex’s formation, more studies such as DSC, PXRD, and SEM measurements were followed.

The crystallinity of PCC2, PCC4, and PCC6 (PPD:CER = 1:4, *w*/*w*) was studied using DSC and PXRD. The melting point of raw PPD was determined to be 185.14 °C. Corresponding endothermic peaks were observed for PPD and the physical mixture but disappeared in PCCs (Figure 4). The thermogram of CER showed a broad endothermic peak at 120 °C [40]. This might be due to the loss of water content retained by the CER, which generally has an intrinsic water content in CER. The endothermic peak of PDD disappeared from PCC, indicating that PPD was converted to an amorphous form [2,41]. The PPD would form an electrolyte complex with CER without PPD-PPD molecular bonding.

To characterize the crystallinity of PCC2, PCC4, and PCC6, the PXRD results were shown (Figure 5). Since PPD was crystalline, it showed sharp peaks in the diffraction angle of 5°–40°, whereas all CERs had reflection peaks of an amorphous nature [3]. The physical mixture and CER, although less intense due to the dilution by CER, remained crystalline and exhibited the main characteristic crystalline peaks of PDD. All the PCCs had no specific diffraction peaks, indicating that the formation of PCC changed the physical state of the drug from crystalline to amorphous. This result was consistent with DSC data and confirmed that the drug molecule was uni-molecularly bound within the CER, indicating an amorphous form [42]. As a result, the DSC and PXRD measurements showed that PCC2, PCC4, and PCC6 were successfully complexed and loaded with PPD using all three CERs.

### 3.2. In Vitro Release Test of PCCs

The release results for raw PPD and PCC1~PCC6 are shown in Figure 6 and Figure 7. In the pH 1.2 buffer (Figure 6), the raw PPD showed a drug release of >90% within 30 min, and the release of PPD in PCCs was suppressed to <30% in the order of PCC1 > PCC3 > PCC5 (Figure 6A) [43]. Among PCC1, PCC3, and PCC5, the release of PPD slightly decreased as the particle size of the CER increased (Figure 6A). However, in PCC2, PCC4, and PCC6, the particle size of the CER did not affect the release of PPD, and the drug release was less than 10% until 120 min (Figure 6B). This finding suggested that the particle size and weight ratio of PPD over CER controlled and suppressed the release of PPD in acidic conditions such as a pH1.2 buffer because there was little cationic ion to exchange PPD molecules. Thus, PPD-CER complexes, when formulated into tablets or capsules, would exhibit a limited PPD release at pH 1.2 despite disintegration in the oral cavity or stomach.

In the pH 6.8 buffer (Figure 7), PCC1 showed an immediate release of >90% within 10 min, which was faster than the release of raw PPD. However, the release of PPD from PCC3 and PCC5 was delayed and sustainable, showing about 93.9 ± 2.6% and 62.0 ± 2.5% at 1440 min, respectively (Figure 7A). The particle size of CER in PCC affected the release of PPD. The small particle size of PCC1 resulted in an immediate release profile because the larger specific surface area of PCC1 and shallow positioning of PPD from the surface rapidly released PPD. However, the larger particle size of PCC3 and PCC5, which had more PPD distribution depth and volume made the release of PPD more sustainable compared to PCC1 [26]. In the weight ratio of PPD over CER (1:2 versus 1:4), the release (%) of PPD was more sustainable with the higher ratios of PCC2, PCC4, and PCC6 (Figure 7B), compared to the lower ratios of PCC1, PCC3, and PCC5, respectively. PCC2 showed a saturated release at 240 min, showing a drug release of 88.8 ± 5.1% at 240 min, and PCC4 and PCC6 showed continuous release until 1440 min, showing a drug release of 88.2 ± 4.8% and 66.0 ± 6.7%, respectively (Figure 7B). The higher weight ratio of CER reduced the release of PPD from PCCs since the rate of dissociation and diffusion of PPD in the PCC was decreased. These facts inferred that the drug was distributed not only on the surface but also inside the CER in PCC. This was also confirmed via SEM, FT-IR, DSC, and PXRD results. When PCC is administered orally, the PPD can be released from an ion exchange resin by replacement with counterions present in the gastrointestinal tract, and the release can be controlled according to the particle size of CER. PCCs with a controlled particle size had the advantage of a simple manufacturing method and no burst effect, even at high PPD loading rates. 

The drug release at 480 min in pH 6.8 of PCC with different mixing ratios of drug and CER with different particle sizes was plotted and shown in Figure 8. This result showed a linear decrease in drug release with increasing particle size. In particular, the correlation was 0.8794 at the 1:2 ratio with a low CER-to-drug ratio, and the correlation was higher than 0.8315 at the 1:4 ratio. Therefore, the particle size of CER suggests that a lower ratio of CER to PPD may have a greater effect on drug release. 

The release results suggested that both immediate and sustained release of PPD could be realized by adjusting the particle size of the CER and the ratio between PPD and CER. Therefore, the desired release pattern can be controlled through the PCC mixture prepared by adjusting the ratio of each PCC that represents immediate release and sustained release. For making up the optimized PPD release profile, a PPD mixture (PCC1:PCC2:PCC6 = 10:60:30, *w*/*w*) was prepared in a gelatin capsule and tested (Figure 9). The target release (%) was intended to be higher than 85% at 720 min with a controlled release pattern. In Figure 8, the calculated release percentage of the PCC mixture was calculated from the average releases (%) of PCC1, PCC2, and PCC5 at each time point in Figure 7A,B, considering the mixing weight ratio of the PCCs in the mixture. As a result, the PCC mixture demonstrated a similarity between the experimental release curve and the calculated release curve. Furthermore, the experimental release of PPD at 720 min was 90.7%, which met the criteria of >85%. The release (%) from each of PPC1, PCC2, and PCC6 was maintained and repeated in the PCC mixture. This strongly suggested that the PCCs in this study can be used to control different release profiles of PPD via a simple mixing PCC. Based on the above results, it was summarized that the CER complex using a controlled particle size is an effective method for developing various drug release controls.

## 4. Conclusions

The purpose of this study was to study the effect of different particle sizes of CER on the formation ability and release characteristics of PPC. PCC was prepared with CER with different particle sizes at the weight ratios of PPD to CER (1:2 and 1:4). In the pH 6.8 buffer, PCC1 prepared with an average particle size of 100 μm CER showed 95% immediate release at 15 min, and PCC4 prepared with an average particle size of 400 μm CER showed sustained release of 80% at 1440 min. FT-IR spectra confirmed possible interactions between drugs and CERs. PXRD and DSC confirmed that the drug was in an amorphous state in PPC. CERs with different particle sizes allow controlled release of PPDs, from immediate release to sustained release. The release of PPD from PCC decreased as the CER particle size increased and the weight ratio of CER increased. The PCC developed in this study can act as an efficient drug delivery system to control the release of PPDs in various ways.

## Figures and Tables

**Figure 1 pharmaceutics-15-00932-f001:**
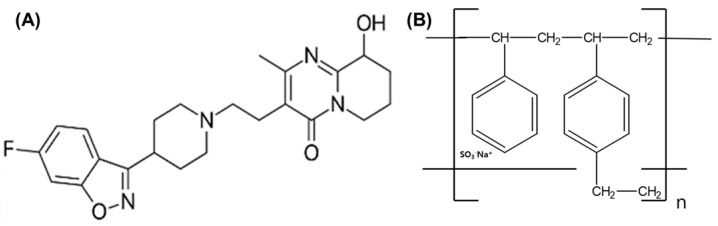
Structure of paliperidone (**A**) and cation-exchange resin (**B**).

**Figure 2 pharmaceutics-15-00932-f002:**
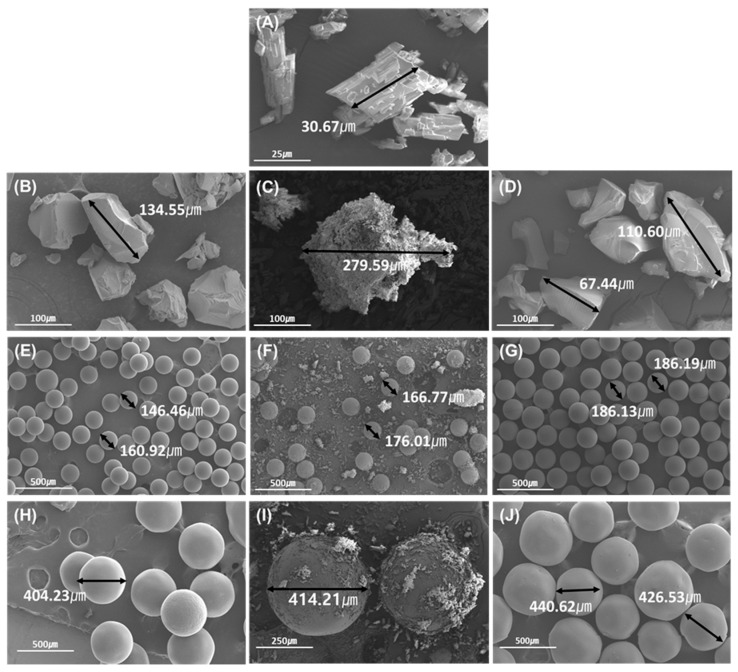
The SEM images of raw PPD (**A**); C1 (**B**); the physical mixture of PPD and C1 (**C**); PCC2 (**D**); C2 (**E**); the physical mixture of PPD and C2 (**F**); PCC4 (**G**); C3 (**H**); the physical mixture of PPD and C3 (**I**); and PCC6 (**J**).

**Figure 3 pharmaceutics-15-00932-f003:**
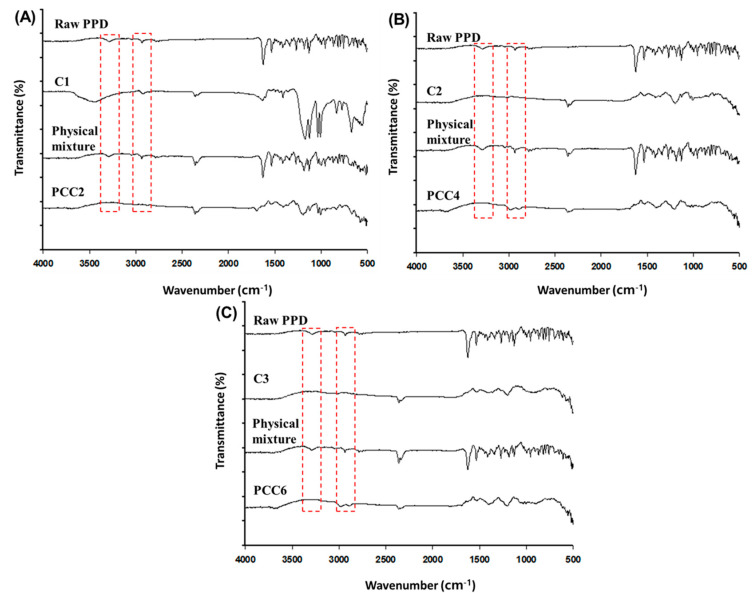
FT-IR spectrum of raw PPD, CER, the physical mixture of PPD and CER (C1, C2, and C3), and PCC; C1 and PCC2 (**A**); C2 and PCC4 (**B**); and C3 and PCC6 (**C**).

**Figure 4 pharmaceutics-15-00932-f004:**
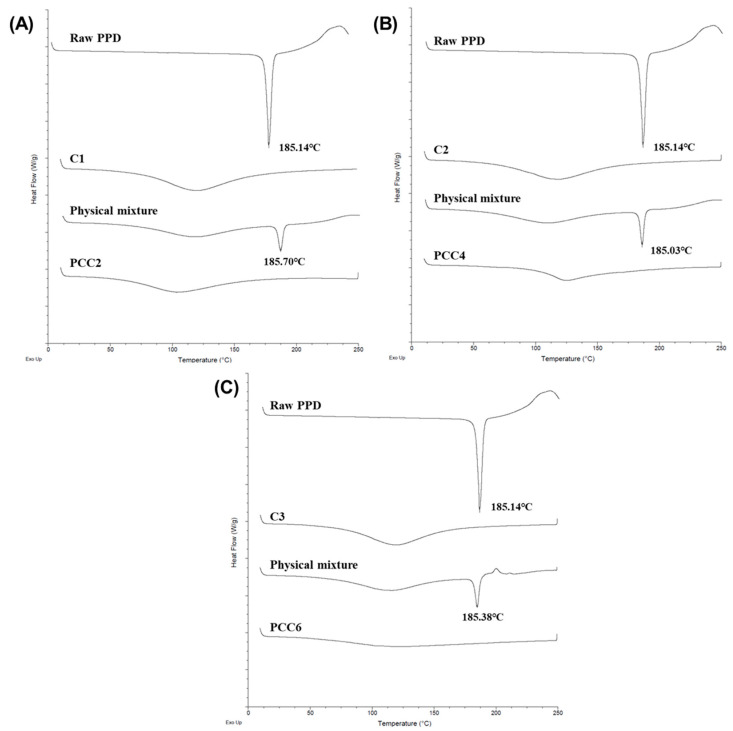
DSC thermogram of raw PPD, CER, and the physical mixture of PPD, CER (C1, C2, and C3) and PCC; C1 and PCC2 (**A**); C2 and PCC4 (**B**); and C3 and PCC6 (**C**).

**Figure 5 pharmaceutics-15-00932-f005:**
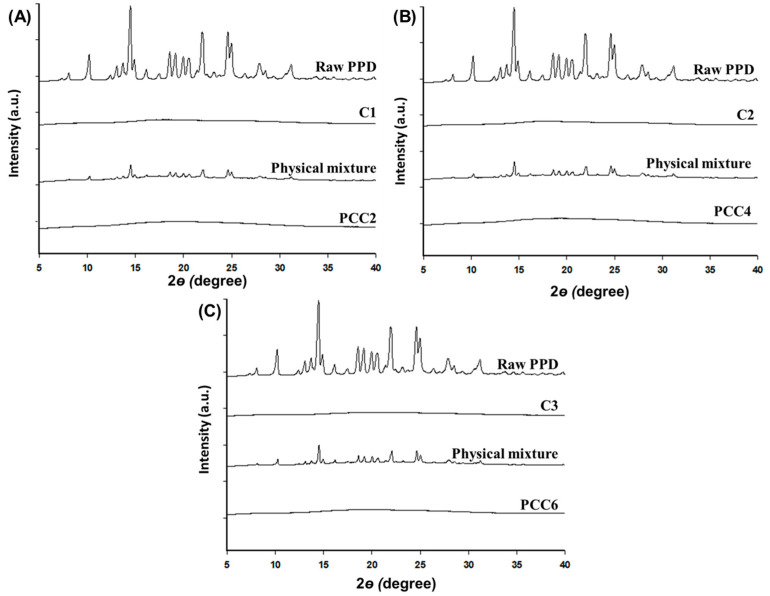
PXRD patterns of PPD, CER (C1–C3), the physical mixture of PPD and CER, and PCC: C1 and PCC2 (**A**); C2 and PCC4 (**B**); and C3 and PCC6 (**C**).

**Figure 6 pharmaceutics-15-00932-f006:**
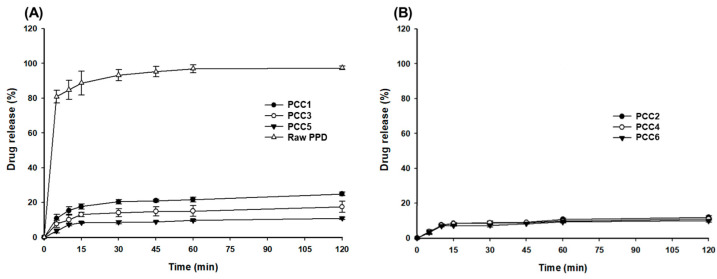
The PPD release profiles at pH 1.2 buffer: raw PPD, PCC1, PCC3, PCC5 (**A**); PCC2, PCC4, PCC6 (**B**).

**Figure 7 pharmaceutics-15-00932-f007:**
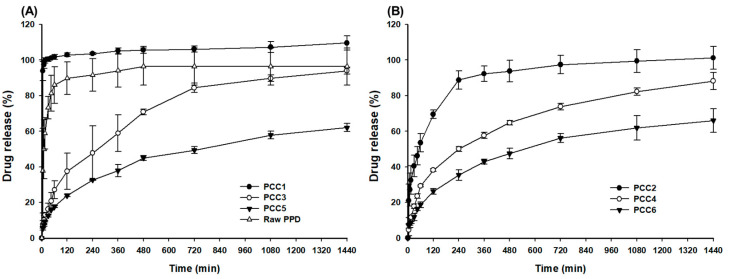
The PPD release profiles at pH 6.8 buffer: raw PPD, PCC1, PCC3, and PCC5 (**A**); PCC2, PCC4, and PCC6 (**B**).

**Figure 8 pharmaceutics-15-00932-f008:**
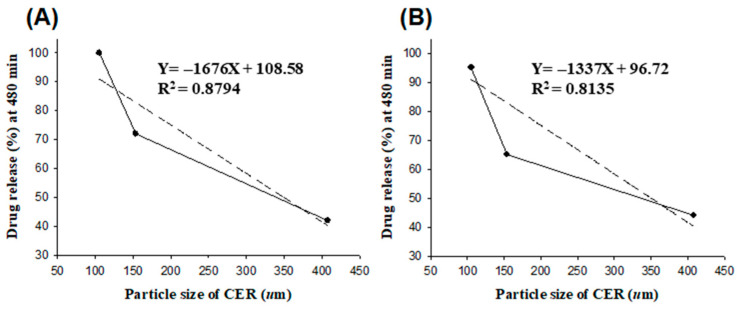
Linear relationship between drug release (%) at 480 min and CER particle size: PPD:CER = 1:2 (**A**); PPD:CER = 1:4 (**B**).

**Figure 9 pharmaceutics-15-00932-f009:**
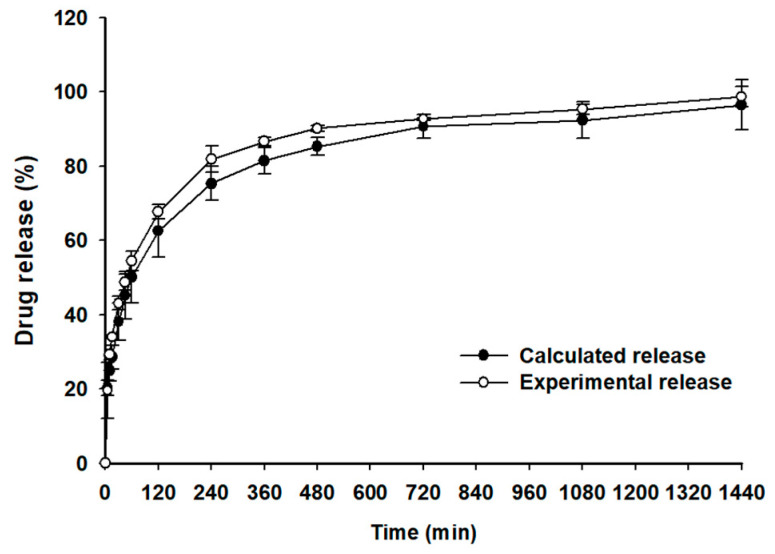
Calculated and experimental PPD release profiles of PCC mixtures (PCC1:PCC2:PCC6 = 10:60:30, *w*/*w*) at pH 6.8 buffer.

**Table 1 pharmaceutics-15-00932-t001:** The properties of CERs (C1–C3).

CER	Shape	* Particle Size (μm)	CER Brand Name
C1	Powder flake	105.5 ± 7.6	Amberite^TM^ IRP69
C2	Spherical bead	153.7 ± 17.2	DIAION^TM^ UBK530
C3	Spherical bead	407.6 ± 20.1	Amberite^TM^ IR69F

* Data are presented as means ± standard deviation (n = 3).

**Table 2 pharmaceutics-15-00932-t002:** The Binding efficiency (%) of PPD in PCCs.

PCC	CER	PPD:CER Ratio	* Binding Efficiency of PPD (%)
PCC1	C1	1:2	99.28 ± 0.02 **
PCC2		1:4	98.86 ± 0.18 *
PCC3	C2	1:2	99.93 ± 0.02 **
PCC4		1:4	99.93 ± 0.10 **
PCC5	C3	1:2	62.10 ± 0.05
PCC6		1:4	98.31 ± 0.13 **

* Data are presented as means ± standard deviation (n = 3). ** *p* < 0.05 compared with PCC5.

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
