# Peer review of "Paliperidone–Cation Exchange Resin Complexes of Different Particle Sizes for Controlled Release"

_pharmaceutics, 2023, doi:10.3390/pharmaceutics15030932_

Round 1
Reviewer 1 Report
Dear Authors,
I like this manuscript very much. It is written very precisely. All the questions I may have, have been explained in the following sections of the manuscript. Congratulations on the promising results. Maybe the only consideration is the application of models to release results. The conclusions are a bit too general, they can be described more in words (less raw data in the conclusions).
Reviewer 2 Report
The manuscript “Paliperidone–Cationic Exchange Resin Complexes of Different 2 Particle Sizes for Controlled Release” presented by Jun‐Pil Jee et al. describes an investigation of different parameters of paliperidone–cation exchange resin complexes (PCCs) and their preparations on paliperidone release from PCC particles in the aqueous phase. Presented results enhance knowledge of physicochemical characteristics of loaded with drugs PCCs and guide the manufacturing process optimization for such drug delivery systems. The manuscript is well written with sufficient factual information provided. However, a number of clarifications should be made for a better understanding of the described work and obtained results.
Additional comments (Comments are made during continuous reading of the manuscript. Therefore, answers to some raised questions may occur later in the text.)
Title
1) Title is cryptic. Release of what? “controlled release” – what control means here?
2) “Cationic Exchange Resin” should be “Cation Exchange Resin”.
3) Something like the following may state the main outcome of the work more directly - “Dependence of paliperidone release on size of Paliperidone–Cationic Exchange Resin particles”
Abstract
1) What does the following mean “raw PPD experienced a complete PPD release”? Define “release”. Define “raw”.
2) Define “controlled release”.
3) How significant was the difference in release rates for different particle sizes?
4) What increase in CER ratio means - a nonproportional increase in load of PPD on larger particles? In this case, a decrease in release rate is not logical for at least for the first minutes or hours. What is the dynamics of the release?
Introduction
1) What are “charged ions”?
2) “insoluble synthetic polymers” – insoluble where or by what?
3) Define “controlled release”.
4) How “the particle size of ion-exchange resins is very important in determining complex formation”?
5) Define “meq/g”.
6) Correct “cationic-exchange”.
7) Previous art on PPD delivery systems based on CER is not well covered e.g. https://patents.google.com/patent/CN105310980A/en and https://www.researchgate.net/publication/312782411_Tailoring_the_Release_of_Drugs_Using_Excipients
Materials and methods
1) Probably a typo “Cu/Kα radiation source”.
2) How were parameters e.g. thickness of gelatin capsule shell controlled?
3) No section on statistical analysis and the number of repeats can be found in the Method section.
Results and discussion
1) First two paragraphs of the Result section belong more to the Introduction section.
2) “significantly high binding efficiency of >98%.” To what? What is the level of significance?
3) Define “binding efficiency (%)”. What is "concentrate of filtrate" in the formula? How was concentrate measured? It is not clear how binding efficiency depends on the concentration of CER. Also, “slightly lower binding efficiency was due to its relatively large particle size and specific small surface area”. Present a formula for the calculation of the binding efficiency which includes these variables. Define “specific small area”. How this relates to “This significant increase of PCC particle size means that the PPD molecules diffused into inner CER and formed electrostatic bonding, rather than the simple surface bonding.”?
4) Not sure what the following means “larger CER particles are difficult to diffuse,”.
5) It is not clear what is measured by transmission FT-IR if probe irradiation can be absorbed (blocked) by solid particles. There is no obvious difference between C3 and PCC6 traces in Fig 2C and C2 and PCC4 traces in Fig. 2 B. Please demonstrate differential traces to visualize observed effects.
6) Define “physical mixture,”.
7) It is not clear why many described results have added references e.g. “peak band in the 238 physical mixture [17, 35].”. Does it mean that the presented data are the same as those published before?
8) Move EM result description in front of other measurements. This will help to understand other results better including the physical mixture sample’s nature.
9) Figure 5. C is not informative. What is depicted – PDD or C1 particle or PDD-coated C1 particle?
10) Describe how to release rate % data are calculated. Fig 6 should show zero release rate at 120 min time point for many experimental conditions, especially PDD. Mention in the text what methods were used for obtaining data for Fig. 6 and 7. Also, “the average release (%) of PCC1, PCC2, and PCC5 at each time points in Figure 7(A)”. Is it rate % or release%? When both should reach zero, at which time point?
11) Add statistical information to support this statement “particle size of CER in PCC significantly”.
12) Present plots demonstrating the dependence of measured parameter(s) on particle size. What is correlation equation(s) – linearity etc.?
Conclusions
1) Only the last three sentences of this section fit the purpose and the rest are just summaries of results and can be removed.
Round 2
Reviewer 2 Report
The authors made significant improvements in the presentation of the results of their work. Just several additional clarifications are needed.
Response 3: I greatly appreciate your valuable feedback. Upon careful consideration, I have decided to retain the current title for the manuscript. However, I have made a slight revision to the title by replacing "cationic" with "cation".
Revised The revised title is as follows: "Paliperidone-Cation Exchange Resin Complexes of Different Particle Sizes for Controlled Release".
Comment: The title ending is not complete. It should be appended with Paliperidone or “the pharmaceutical”.
Point 7. Abstract - 4) What increase in CER ratio means - a nonproportional increase in load of PPD on larger particles? In this case, a decrease in release rate is not logical for at least for the first minutes or hours. What is the dynamics of the release?
Response 7: Thanks for the good point. Revised according to the reviewer's opinion.
Added “The drug release at 480 min --- have a greater effect on drug release.” and “Figure 8” in line 348-353 and line 354-356, respectively.
Comments: How release can affect the same release in the following “The drug release at 480 min --- have a greater effect on drug release.”? Probably --- mean something.
Point 12. Introduction - 5) Define “meq/g”.
Response 12: Thanks for the good point. Revised according to the reviewer's opinion.
Revised “In particular, as a representative CER, --- of methacrylic acid and divinylbenzene [23].” in line 92-94.
Comment: The response is not clear. It looks like, the authors want the reviewer to go to the main text, look for a particular sentence and verify their response. Sorry, it is too time-consuming approach for both parties.
Point 21. Results and discussion - 4) Not sure what the following means “larger CER particles are difficult to diffuse,”
Response 21: Thank you for your valuable feedback. Larger CER particles have a smaller specific surface area, resulting in a lower binding efficiency for PPD due to a smaller surface area available for binding. As a result, PPD molecules are likely to diffuse into the interior of the CER particles for better binding and complex formation.
Revised “Furthermore, when PPD molecules react with larger CER particles, they are more likely to diffuse into the interior of the particles, which reduces mass transfer and makes most of the inner resin bead less accessible for binding. This results in a lower binding efficiency compared to smaller CER particles with the larger specific surface area” in lines 237-241.
Comment: To clarify the meaning of conveyed information, the authors need to add schematics illustrating a) PDD diffusion into the interior of the particles; b) specific surface area; c) PDD interaction with specific surface area; d) the inner resin bead; c) how diffusion of PDD into the interior of the particles makes most of the inner resin bead less accessible for binding. Without such schematics, the description is very difficult to understand.
Point 22. Results and discussion - 5) It is not clear what is measured by transmission FT-IR if probe irradiation can be absorbed (blocked) by solid particles. There is no obvious difference between C3 and PCC6 traces in Fig 2C and C2 and PCC4 traces in Fig. 2 B. Please demonstrate differential traces to visualize observed effects.
Response 22: Thank you for bringing up this point. After careful review of Figure 2, a new additional difference between PCC and CER was found in the peak range of 3200 cm-1 to 3400 cm-1. Although this FT-IR result alone is incomplete to confirm the formation of electrostatic bonding, it is one of the supportive data along with DSC, XRD, and SEM results. Taken together, these results clearly indicate the formation of the complex.
Revised “In the raw PPD and the physical mixture of PPD and CER, characteristic peak bands of PPD were observed at 2800 cm-1 ~ 3000 cm-1 and 3200 cm-1 ~ 3400 cm-1. However, in PCC, these two peaks were not present, as indicated by the red-dotted square in Figure 3.” In lines 270-273.
Comment: Answer to the questions is not complete. What transmission FT-IR analyzes – entire particle, its core, or just solution surrounding particles? Is it any concentration dependence in results (if such measurements conducted)? Please present an overlay of traces for regions of interests at higher magnification.
